# Navigating Uncertainties: How to Assess Welfare and Harm in Genetically Altered Animals Responsibly—A Practical Guideline

**DOI:** 10.3390/ani10050857

**Published:** 2020-05-15

**Authors:** Anne Zintzsch, Elena Noe, Herwig Grimm

**Affiliations:** 1Institute of Laboratory Animal Science, University of Veterinary Medicine Vienna, 1210 Vienna, Austria; 2Messerli Research Institute, University of Veterinary Medicine Vienna, Medical University Vienna, University Vienna, 1010 Vienna, Austria; elena.noe@vetmeduni.ac.at (E.N.); herwig.grimm@vetmeduni.ac.at (H.G.)

**Keywords:** genetically altered animals, welfare assessment, uncertainty, severity classification, harm-benefit analysis, 3RsAGENT

## Abstract

**Simple Summary:**

When using animals in research, ethical aspects must be included in the project evaluation process. As one important part, a harm–benefit analysis (HBA) should be carried out in order to approve projects in line with the EU Directive 2010/63/EU, which sets out the rules for animal experiments in Europe. These regulations state that the harms and benefits of a project should be assessed and weighed before the project starts. Assessment of harms caused by scientific procedure is a precondition for ethical evaluation. In this context, projects that involve genetically altered (GA) lines raise new issues. A significant lack of knowledge surrounds new GA lines, making it difficult and sometimes impossible to estimate harm prospectively with sufficient certainty since it is not predictable what sort of harm—if at all—the animals are going to experience. Therefore, this contribution aims to deal with the challenges of harm assessment in GA animals and their implications for welfare assessment and the HBA. A practical guideline is presented herein to serve as guidance for relevant harm factors and address the main challenges, particularly when dealing with uncertainties in project evaluation.

**Abstract:**

The use of animals in research requires careful ethical consideration of whether the burden on the animals is justified. As one important part of the project evaluation, a harm–benefit analysis (HBA) must be carried out in order to approve projects in line with the EU Directive 2010/63/EU. This implies that harms and benefits must be assessed prospectively beforehand in order to weigh them. Although there are different methods of weighing, it is clear that an assessment of prospective harms and benefits is a precondition for any weighing procedure. In this context, projects that use genetically altered (GA) lines raise new issues. A unique challenge when using GA lines is the significant lack of knowledge in this context, making it difficult and sometimes impossible to estimate harm prospectively with sufficient certainty, since it is not predictable what sort of harm—if at all—the animals are going to experience. Therefore, this contribution aims to deal with the challenges of harm assessment in GA animals and their implications for welfare assessment and the HBA. A practical guideline is presented herein to serve as guidance for relevant harm factors and address the main challenges, particularly when dealing with uncertainties in the process of HBA.

## 1. Introduction

Genetically altered (GA) animals are widely used in biomedical research and are valuable models in the study particular gene functions or when investigating disease mechanisms. Advanced gene-editing technologies like CRISPR/Cas9 further extend research areas of application [1,2], and it is reasonable to assume that GA animals will continue to have a prominent place in biomedical research. In February 2020, a compilation of European data was published under the new requirements of Directive 2010/63/EU (Dir.). In 2017, 74% of animals used for the maintenance of GA lines were reported as GA animals without a harmful phenotype and 20% with a harmful phenotype [3]. GA mice, rats, and fish are among the species most commonly used for scientific purposes. For example, in Germany, mice are the most common species (68%) used, with 53% of mice being GA, of which 22% are GA with a harmful phenotype [4]. However, the collection and processing of data on GA animals across Europe require coordinated regulations to ensure transparency regarding the animals used in biomedical research and to inform the public accordingly.

According to Article 3 of Directive 2010/63/EU, breeding animals with a likely harmful phenotype is subject to a project evaluation. The directive provides a framework and sets out the conditions and components of project authorization in Article 38. GA animals are subject to approval if they are used in harmful procedures, which explicitly encompasses the breeding of animals with a likely harmful phenotype. An integral part of each project evaluation is a severity assessment and the consequent classification of pain, suffering, distress, and lasting harm that factors into a harm–benefit analysis (HBA). Providing information about the nature and severity of the animals’ harm, pain, suffering, and distress is a legal requirement and serves as an important aspect of the project evaluation. Nevertheless, authorities and applicants still face the challenge of fulfilling this legal obligation and providing sufficient information on projects of GA lines that are likely to develop a harmful phenotype.

Smith and Jennings (2004) [5] noted that “it is particularly difficult to assign severity categories when adverse effects are uncertain or unpredictable, such as in the production of genetically modified animals or in toxicity testing”. Bert et al. (2016) [6] recently highlighted the difficulties that come with the implementation of the Dir. as “there is a large gap between the legal obligation to ensure animal welfare and the lack of objective biomedical indicators for measuring animal well-being”. This is specifically true with regard to GA animals, since a systematic account on how to deal with these uncertainties is still lacking. In this article we aim to provide guidance for navigating challenges related to harm assessment in GA animals when little to no empirical knowledge is available. We argue that without prior knowledge of the expected phenotype, harm assessment is not plausible. This should be made transparent and a severity classification on a highly speculative basis should be avoided. To deal with uncertainty responsibly, the planning of monitoring intervals, harm assessment, and identifying humane endpoints should become the main tasks.

In addition to the phenotypic characteristics, all related procedures for the generation of a new line must be taken into account. In the past, some Ethics Committees disregarded the associated procedures to generate animals with a desired phenotype [7], but with the Dir. and the reporting requirements [8], e.g., superovulated females, vasectomized males, and embryo-transplanted surrogate mothers, are explicitly mentioned. The legal requirement to evaluate GA animals applies to new lines that are generated by genetic engineering and by cross-breeding established lines, as well as established lines that are likely to develop a harmful phenotype. In these cases, a full project application, including all related procedures and consistent evaluations of each factor according to Article 38 of the Dir., is required. It is therefore important to note that all interventions and conditions must be considered to ensure animal welfare at all stages of life and to implement refined procedures as early as possible [9]. All these aspects are considered in the present work.

Although GA animals are a vital part of biomedical research, the implementation of special features of GA animals in the project evaluation and especially in the context of the HBA is not the focus of published literature. Taking up the challenge, we developed a practical guideline for the use of GA animals in scientific procedures and herein provide a hands-on tool, the 3RsAGENT, to document the relevant information for the project authorization.

## 2. GA Animals in the Project Evaluation: Facing Uncertainties

### 2.1. Harm Assessment in the Context of the HBA

The project evaluation (Directive Art. 38) is divided into several steps carried out by the competent authority. Naturally, harm done to animals is a key factor in this process and the assessment and classification of harm reported to the authority plays a crucial role. In fact, without a solid harm assessment provided to the authority, the HBA, as one part of the project evaluation, cannot be carried out plausibly. However, not only the prospective severity assessment but also the actual severity assessment during the project and a retrospective assessment from which future projects will benefit are part of the harm assessment [10]. If the severity of a procedure changes, it is imperative to re-evaluate the balance of harm and benefits. For this reason, expected harm and foreseeable and unexpected harm add to the quality of the project and must be taken into account. 

Weighing harm against benefits requires transparent and systematic collection of information of the expected harms and benefits, which are the basis for the HBA [11]. All factors causing harm to animals must be collected and the severity of harm must be classified. A clear picture regarding all harms is the prerequisite for the severity classification of the project (Directive Art. 15). The prospective severity classification, which is directed toward the animal suffering the most, should not be confused with the total overview of harm done to animals in the project, which is the benchmark for harm in the HBA. Going beyond the severity classification, the systematic description of all harm involved in the project is a precondition for the HBA (total overview of harm), whereas the overall severity classification of a project focuses on the individual suffering the most (individual most harmed). The balance of harms and benefits and approval that the harms and benefits are in proportion (proportionality) is only legitimate if three preconditions are in place: (a) A legal scientific objective (legitimacy); (b) the suitability of the project design (suitability); (c) the impossibility to achieve the objective with less harm and/or milder means (necessity) [12]. Only on this basis is it plausible to carry out the HBA in order to check whether the proportionality condition is also met. The challenges of defining the benefits of a project were described previously in the literature (e.g., [13,14] and are not discussed in this work.

### 2.2. The Concept of “Harm”

When it comes to the HBA and the assessment of harm as its precondition, it is often challenging to identify relevant harm dimensions. In general, phenotypic characteristics and all effects caused by generating, breeding, or maintaining GA animals are of relevance for the assessment and severity classification. However, the assessment of a *likely harmful phenotype* triggers debate and is in need of clarification. Therefore, the idea of “lasting harm” due to a harmful phenotype is addressed in more detail herein.

Although genetic alterations of animals are considered as harms in the light of respect to naturalness [15], they do not per se have an effect on animal welfare [16]. Hence, two concepts of harm are relevant in this context, i.e., *subjective* and *objective* harm. Whereas subjective harm is to be understood as a negative subjective feeling or condition that is experienced by the animal, objective harm is to be understood as negative influence on an animal without negative subjective experience [17]. Negative effects on the animal’s experienced welfare and mental condition, such as behavioural restriction, pain, suffering, and distress, are relevant parameters [18] for subjective harm. Contrary to subjective harm, objective harm plays an important role in GA animals. Such harms are not experienced by the animal per se, such as genotypic modification [16]. Consequently, harm that can be identified from a third person’s perspective, but which is not experienced by the animal is termed “objective harm” in the following. In this regard, the modification of the genotype might be considered harmful to the animal in objective terms, since the species-specific set of genes is altered. Only the concept of subjective harm, but not objective harm, found its way into the Dir. Relevant harm that must be considered in the project evaluation is defined in terms of suffering, pain, and distress in Art 38: “a harm-benefit analysis of the project [must be carried out] to assess whether the harm to the animals in terms of suffering, pain and distress” is taken into consideration. This is also in line with published literature, e.g., the Nuffield Council on Bioethics (2005) [19], with the focus lying on these welfare implications.

Hence, within the scope of this work to develop practical guidelines, the harm dimension should be understood accordingly: Following the Dir., a procedure that causes a level of pain, suffering, distress, or lasting harm (harm criterion) equivalent to or higher than that caused by the introduction of a needle (needle criterion) in accordance with good veterinary practice is considered relevant harm in the project evaluation (Art. 38). Only if it can reasonably be argued that a genetic modification leads to negative subjective experience in an animal is it harm that can and must be taken into account. In consequence, objective harm can, for legal reasons, not influence the positive or negative outcome of the project evaluation. This is in line with Dahl et al. who classify lowered infertility as not causing welfare problems from a hedonistic point of view [20]. In contrast, a Danish study included decreased fertility as a factor of harm [21]. Recently, Brønstad and colleagues compiled a list of harm factors as animal welfare harm to represent a collection of subjective and negative experienced harm factors on which the current work is based, including pain, injury, or disease and fear, anxiety, or distress [11]. 

### 2.3. Severity Assessment and Classification of GA Lines 

Aside from the differing views on what determines a harm factor in GA animals, general rules regarding severity assessment also apply to these projects. As a first step of a project review, Article 15 of the Dir. requires a prospective severity assessment that includes each procedure in a project and results in a severity classification. Further, a severity degree (non-recovery, mild, moderate, severe) must be assigned to the entire project on the basis of the most severely harmed animal [10]. Beyond this legal requirement to classify the severity, an actual severity assessment during the project ensures continuous adaptation of the protocol, and a retrospective assessment provides the opportunity to summarize findings and to gain valuable information for projects in the future. Guidance on how to assign severity can be found in Annex VIII of the directive or in several working group reports and recommendations [22,23,24,25,26,27]. Elements in an individual severity assessment include the type of technique, species, strain, stage of development, previous experience, frequency, intensity, duration of effect, and effectiveness of stress-reducing measures. When assigning an overall severity for the animal’s experience over the lifetime, cumulative effects of each intervention should be carefully reviewed and an overall severity degree assigned [10]. In case of GA lines, it is important to note that different animals of a line can develop divergent genotype-related phenotypes or undergo different procedures and therefore can differ in their overall experienced severity of harm. The proportion of animals in different severity categories should be indicated separately, as the number of animals experiencing various degrees of harm influences the total harm considered in the HBA. 

For reporting reasons, only procedural harm is required to be taken into account when reporting a severity degree for individual animals, implying that only adverse phenotypes related to the genetic alteration should be included in the assignment of severity in GA animals. Background-specific characteristics (even if these lead to the experience of pain, suffering, or distress) must be disregarded according to the requirements of the annual reporting to the responsible authorities [24,28]. Regardless of the reporting rules and the origin of pain, suffering, or distress, the moral obligation to minimize any harm remains unaffected and refinement should be implemented whenever possible. Notably, the severity assessment does not only mean putting labels on procedures; it is more important to carefully describe and evaluate each potential harm factor (see [5]). In the case of GA animals, it is a matter of animal welfare that this obligation also applies to any adverse effects resulting from the genetic background strain, even if not covered by the statistics or the project authorization.

### 2.4. Risk, Probability, and The Factor of Uncertainty in GA Animal Projects

The prospective harm assessment of animals of a new GA line can be compared to a general risk assessment and therefore called a welfare risk assessment [20]. Speaking in terms of a risk assessment and according to Dahl and colleagues, a preliminary evaluation of welfare issues is based on the identified hazards (here: a harmful phenotype and invasive procedures during generation) and, according to the identified risks, an appropriate strategy for further investigation (here: a systematic welfare assessment of the animals) must be developed.

Brønstad et al. (2016) [11] postulated that “harm has also been defined as a product of the probability and the severity of harm” and refers to The Research Ethics Guidebook by Boddy et al. [29], which is available online. This definition applies to procedures where existing knowledge, e.g., regarding the efficiency of the genetic engineering method or Mendelian ratio for the distribution of genotypes, can be used. However, in cases where the effect is unknown, probability cannot be used as parameter to describe the likelihood and the proportion of animals that might experience a defined severity of harm. 

In effect, this practically ruins the very foundation of an HBA, in which one of the preconditions [30] is the assessment of harm. With this in mind, Schuppli (2004) [15] also claimed that unpredictability of phenotypes in new lines makes it virtually impossible for Animal Ethics Committees or authorities to judge the level of harm relative to the expected benefits. It was further argued that it is difficult to aim for refinement prospectively, and in these cases “careful attention to monitoring and endpoints are the obvious options”. 

For projects with high uncertainty, similar points apply. If no informed decision regarding the severity of the phenotype or the procedure is possible, it should instead be asked which adaption of the project design (here: monitoring and welfare assessment) is necessary to fulfil the requirement of the best possible refinement. With respect to a new GA line, a systematic welfare assessment of the animals over their entire lifespan is the suggested strategy [31]. Time points and frequency of monitoring should be adapted to the expected phenotype to ensure appropriate care of the animals. Established monitoring systems and line-specific assessment sheets include symptoms or thresholds that support the determination of tailored refinement strategies. If established lines are introduced to a new animal facility, continued monitoring in the sense of an ongoing actual welfare assessment ensures the assessment of the phenotype under local conditions. To ascertain the quality of science and the use of standardized disease models, continued monitoring of the genetic background and the quality of genetic alteration should be a matter of course and is in line with good scientific practice [32,33,34]. If an unexpected phenotype is observed, the line should be re-evaluated on the basis of a systematic welfare assessment [31]. A thorough retrospective evaluation complements the evaluation under conditions of uncertainty.

Nevertheless, applicants and competent authorities are still challenged with a prospective severity classification on the basis of little to no empirical evidence when new lines are established. Especially in cases where the severity of the induced phenotype is the main scientific question, researchers hesitate to set early endpoints that would counteract their explorative approach of the new phenotype. At the same time, the lack of knowledge impedes a definition of line-specific early endpoints, making the application of humane endpoints the only option. When new lines are established, the question emerges of whether a meaningful prospective harm assessment of phenotypes is possible at all. Therefore, we suggest making this issue explicit in the application procedure in order to avoid giving the misleading impression that harm could be identified in the HBA without prior knowledge. 

For new lines in which the gene function is known or there are already lines with similar genetic alterations available, this knowledge should be used to describe the expected phenotype. 

For established lines, the phenotype can be described in most cases and, apart from unpredictable events, such as spontaneous mutations and possible impacts on the phenotype, a severity classification can be given on reasonable grounds. This approach follows the same principles as other animal experimental intervention, allowing a severity degree to be assigned.

### 2.5. Considering Different Levels of Available Knowledge in The Course of Severity Assessment

When assessing a GA line, it is important to distinguish between (a) generating a new GA line and (b) breeding and maintaining an established line, especially with regard to given knowledge. Whereas in the latter case existing knowledge regarding harm is available, the former must be differentiated further between developing new GA lines without pre-existing knowledge of the phenotype and engineering or breeding GA lines with an expected phenotype based on knowledge of the gene function or a medical condition of an existing line with similar gene modifications that are of relevance in regard to harm. Whereas the latter assessment builds on existing knowledge, the assessment of the former carries even greater uncertainty. This uncertainty decreases if breeding of an established line is continued and the severity of the expected phenotype is assessed using living animals. The two classes (a and b) differ with regard to available information, on which the severity classification and the consequent harm–benefit analysis must be carried out. 

Respectively, uncertainties of the phenotypic characteristic of a strain may change during a project, emphasizing the need for re-evaluation of a project according to the knowledge gained. For most new lines, the prospective assessment of the phenotype expression is accompanied by a high level of uncertainty. The function and pathophysiological interdependencies are still unknown for many genes [35] and, in these cases, it is obvious that no description of the expected phenotype can be made and a severity classification is not plausible. Only a systematic welfare assessment can give information regarding the expression and possible negative impacts on the animals. In conclusion, the severity of the project is determined by setting general (humane) endpoints. 

If the function of the gene is already known or the genetic alteration of a new line suggests similarity to already existing lines, this information can be used to hypothesize a severity degree. Correspondingly, if knowledge about the expected phenotype is available for established lines, a severity degree, including efficient refinement measures, can be identified prospectively. Both cases enable a description of the phenotype based on a systematic welfare assessment of unknown or unexpected phenotypes, allowing the assignment of an actual severity and serving as the basis for the decision on further breeding. In addition, uncertainties must be indicated and consequences pointed out in order to address them adequately in terms of animal welfare. It may also be necessary to adapt the prospective severity classification to findings during the welfare assessment and, in case of a higher-than-expected severity, a re-evaluation of the project may be necessary.

## 3. How to Deal with Uncertainties?

### 3.1. Assigning An Uncertainty Factor

In order to answer to the question of how to deal with uncertainties, a closer look into the dimensions of uncertainty is in order. In the case of potential harm to GA animals, we face three aspects of uncertainty regarding: (a)The kind of harm;(b)The severity of harm;(c)The probability of harm.

Within the framework of the directive, the three dimensions factor differently into the problem of harm assessment. First, since “harm” is defined as “pain, suffering, and distress” in the directive, the kind of harm that plays a role is clear. As mentioned in Section 2.2, only subjective harm experienced by the animal is of significance when carrying out the HBA. Second, although knowing what kind of harm is significant in the harm assessment, its severity can be uncertain. The guidance provided by the directive in this respect is the severity classification. However, the degree of uncertainty in prospective severity classifications can vary due to existing knowledge. In other words, uncertainty related to the severity classification varies with the availability of relevant knowledge. If extensive knowledge is available, a severity classification is plausible. When only little or non-comparable cases can be found in the literature, the uncertainty increases up to the point where the severity classification is pure speculation. Third, and analogously, the probability of harm can only be plausibly assessed if the given body of knowledge provides reference points.

These three dimensions factor into harm assessment “under uncertainty,” meaning that relevant knowledge is only partially available or unavailable. As mirrored in Table 1, uncertainty can therefore be low, medium, or high depending on the available knowledge. Since uncertainty can be translated into a lack of knowledge and can be reduced with additional knowledge, gaining and sharing experience and publishing results of retrospective harm assessment is of the utmost importance. These concepts are integrated in Table 1.

Prospective assessment always goes along with some degree of uncertainty. The crucial problem, however, is how to act if no relevant probabilities are on hand to support a plausible severity classification. In other words, how should we proceed if no knowledge is available to substantiate prospective harm? One strategy is to follow a probabilistic account (probabilism) and opt for a classification that seems most plausible to the subject taking the decision. In the context of project evaluation, an authority must evaluate the information provided by the applicant transparently. Opting for a subjective decision is not viable. since the severity classification would turn into an arbitrary choice. This would imply that the authority must either believe or disbelieve the applicant without reference to a shared knowledge basis. A second strategy is to always opt for the worst (tutiorism). In this context, this would turn into classifying all animals as “severe” without evidence. This position would not be plausible since, although knowledge regarding the particular project is not at hand, this argument typically applies if the kind of harm is uncertain. For instance, if we considered what would happen if we let GA animals out into the wild with no knowledge of what to expect, it is plausible to opt for the worst case in risk assessment. However, in our case, the kind of harm we are dealing with is not unclear and the number of potentially harmed animals is controlled. Therefore, the tutiorist argument is misguided in this context.

Consequently, we believe this is the most plausible strategy to make the uncertainty explicit and understand the precautionary principle in a way where potential harms are immediately spotted and dealt with as soon as they occur. From this, it follows that the greater the expected uncertainty, the greater the responsibility to make sure that harms are detected by a welfare assessment as soon as possible by qualified persons (see Figure 1).

### 3.2. Consequences of Uncertainty

Due to the immanent uncertainty factors with respect to new GA lines, a transparent way of
(a)indicating uncertainties,(b)adapting the monitoring and assessment scheme accordingly to ensure the recognition of any impairment of animal well-being, and(c)setting (general) humane endpoints
might be the only way forward.

By assumption, the prospective severity degree of the project can only be determined by the definition of endpoints rather than the speculative basis of assigning a severity degree to a new genotype-related phenotype. Table 2 provides the factors of a welfare assessment that should be tailored to the uncertainty factor to ensure the detection of animals with impairments as soon as possible and to draw professionally sound conclusions from the findings.

## 4. The 3RsAGENT as a Tool for Managing Uncertainties

Since prospective harm plays a major role in the HBA as a main aspect of the project evaluation, a structured account on how to provide and evaluate information on harm of GA animals is of significance for applicants and authorities. As outlined previously, assigning an uncertainty factor facilitates transparency and illustrates the need for specific monitoring and defining endpoints on phenotype- or procedure-specific bases or on the basis of a more general severity limit. As the main outcome, we provide a template that could be used to evaluate each step or procedure and give recommendations on consequences for the design of an ongoing severity assessment (Figure 2). The 3Rs Assessment of Genetically Altered Animals Tool (3RsAGENT) guides the evaluation and facilitates a review of chosen methods with respect to defining and best reducing uncertainty or addressing this issue with a caring attitude. In cases where we do not know the outcome, this should be made transparent. A transparent presentation of the uncertainties involved should also be appreciated by the authorities.

The 3RsAGENT provides practical guidance and serves as a step toward harmonizing and standardizing harm assessment in GA animals in the context of an HBA. On one hand, applicants should be provided with transparent criteria and standards that should be addressed in their proposals. On the other hand, the same criteria and standards should be the basis of the authorities’ evaluation. In order to provide a solid basis to interlink the development of applications with the evaluation procedure on the basis of a clear methodology and criteria, we herein identify factors to be included into the assessment of GA animals and explain how these factors can be considered in the project evaluation process. Smith and Jennings (2005) [5] noted that “it is vital that each protocol is described in such a way that it is possible to get a clear sense of the effects, including cumulative or sequential effects (and not just clinical signs), that each individual animal is likely to experience.” 

The 3RsAGENT was designed to provide a comprehensive overview on all welfare relevant aspects when using GA animals in research projects. Users select the components that are relevant to their projects and fill in the table accordingly. The components differ when generating a new line, maintaining a line with a harmful phenotype, or using GA animals in experimental protocols. 

More information on the components of the 3RsAGENT (Figure 2) and criteria of harm alongside facilitating questions can be found in the Appendix A. It is intended as a reminder and should assist with “asking the right questions” to encourage a review of established techniques and standard procedures in the light of the 3Rs. However, it cannot replace a literature review on state-of-the-art of techniques and procedures. The template of the 3RsAGENT is also available online [36]. 

The 3RsAGENT in connection with Table 1 (uncertainties) and Table 2 (consequences) is intended for use during the project evaluation process and can be handed alongside the application. The Appendix A (3RsAGENT: Appendix A and practical guidance) can also be used for a protocol review during the project or retrospectively in decisions regarding such aspects as further breeding of a new line. To illustrate the idea of the tool, two practical examples are provided in the Appendix A.

The 3RsAGENT provides the opportunity to describe and evaluate factors of harm in a transparent manner and serves as a reminder for factors that can be easily overseen. Further management measures (e.g., structured welfare assessment, retrospective evaluation) must be presented on the basis of the uncertainty factor. Whether the uncertainty is adequately managed is an assessment criterion for authorities. With the 3RsAGENT, it should be possible to examine whether, in accordance with the principle of proportionality, uncertainty is dealt with in such a way that potentially occurring harm can be minimized instantaneously. 

## 5. Discussion

### 5.1. Inconsistency in Severity Classification

Severity classification is not yet harmonized across Europe. For example, results of a survey with in total 358 participants uncovered huge differences in classifying the impairment of blind mice [37]. The survey was undertaken during lectures at international scientific conferences and meetings in four different European countries, where participants were asked to vote by an internet poll system for a severity degree. In the mentioned case of blind mice, 50% decided mild severity, 21% voted for moderate, and 13% assigned blindness in mice as severe. The remaining 16% of participants considered blind mice to be a non-harmful phenotype. These results show that no uniform implementation of severity classification has yet been achieved, not even for the question of whether a phenotype is harmful or not. 

Several studies investigated the reasons for these differences. Experience and knowledge about species affect views on the perception of pain [38], and therefore the severity classification of painful procedures. Nationality seems also to affect the attitude to animals and views on the sentience of different animal species, as well as the use of animals in research [39]. Phillipps and McCulloch also demonstrated that gender plays a role, with females appearing to have greater concerns about animal welfare than males. 

Another reason for underestimating harm can be seen in the bias toward avoiding hurdles in the authorization process. If the procedure is categorized into a higher severity category, many scientists worry about increased bureaucracy and longer processing times of their project by the responsible authorities. This may also apply to the obligation for retrospective assessment of severe procedures, even though this evaluation is beneficial for their future experiments. In other cases, we assume that a simple lack of information is to blame, as details are needed to be able to classify phenotypes appropriately. 

In 1994, a FELASA working group noted that pain, suffering, and distress were difficult concepts to apply to laboratory animals and demanded a simple means of accurately grading that could be applied to a wide range of circumstances and procedures [40]. Up until the present time, the same system with its difficulties and challenges seems to still exist, therefore warranting further improvement.

Nevertheless, a grading of harm is helpful to draw attention to more severe procedures, raising awareness of procedural impacts on animals, and informing the public transparently. This becomes particularly pressing under the directive’s requirement to carry out an HBA. If it turns out that the assessment of harm cannot be brought under a feasible and coherent methodology leading to comparable results in comparable projects, the question emerges of whether the HBA turns into vague speculation or an arbitrary venture. This in turn questions the framework of the HBA. In summary, more research and discussions are needed to define more and specific criteria for classification and to avoid subjective evaluations whenever possible.

### 5.2. Documentation and Data-Sharing to Improve the Welfare of GA Animals

Sharing of data is crucial to reduce the uncertainty factor of the genotype–phenotype relationship and to ensure sufficient implementation of refinement in the breeding and maintenance of GA animals. A valuable source of information is provided in online databases, such as the International Mouse Strain Resource (IMSR) [41], Mouse Genome Informatics (MGI) [42], The International Mouse Phenotyping Consortium (IMPC) [43], and The European Mouse Mutant Archive (EMMA) [44]. Even though phenotype characteristics can be found on these respective websites, information regarding welfare-relevant aspects is scarce and, in general, is not included in the description. Therefore, it is of the utmost importance to provide this information when transferring a line to another research institute, including the appropriate designation of the line following international nomenclature rules [45]. National guidelines are already available, e.g., from the German National Committee [46,47], the Swiss Federal Food Safety and Veterinary Office [25], and the British Royal Society for the Prevention of Cruelty to Animals [48]. Nevertheless, in practice, it is not yet evident that information is provided in each case. This approach should be further strengthened to promote the 3Rs while providing sufficient strain information for the scientific purpose.

### 5.3. Challenges of HBA and the Specific Features of GA Animals in This Context 

Although a number of challenges of the HBA were described in the literature and ways to deal with them were identified [11,30,49], the core problem of harm assessment in breeding GA animals was not previously addressed. As we argued, necessary knowledge to assess prospective harm is sometimes lacking in the strict sense. Under such circumstances, measures should be taken on a practical level to reduce potential harm. However, the question remains of how to carry out a (legally required) HBA if nothing or little can be known about the harm threatening the GA animals. Two things can be said in this regard. First, the presented 3RsAGENT aims to assist applicants and authorities alike in navigating in this area. Its purpose is to provide a systematic account for finding out whether, and in which respect, harm assessment in GA lines is at risk of turning into pure speculation and how to deal with potential harm responsibly if some knowledge is at hand. This should enable the applicant to transparently identify uncertainties and measures to deal with them as a part of the project design and preparation for the HBA. Second, if no knowledge of potential harm can be provided, the HBA turns into an obscure venture. The suggested path to providing measures to reduce occurring harm seems to be one plausible way forward, leaving the problem of the HBA unsolved. Since all projects must undergo an HBA, breeding new lines without relevant knowledge of potential harm remains a legally obligatory “mission impossible”. As we argued, we think it is better to face and address this fact transparently in project applications, since the alternative would open the door to unreasonable decisions. 

## 6. Conclusions

Current challenges with regard to harm assessment and severity classification were outlined in this work, indicating that the problem reaches deeper into the foundations and aims of the Directive. With the 3RsAGENT, we aim to provide a step-by-step guide for scientists, Animal Welfare Officers (AWOs), and authorities to identify uncertainties and their consequences, and to provide a tool supporting transparency in the application process. All relevant knowledge should be utilized and shared in order to build one’s assessment on reasonable grounds, wherever possible. Where this is not possible, the precautionary principle should guide the harm assessment and, in cases where such information is not available, transparent communication is needed. The basis for a project evaluation is then formed by the appropriateness of the planned measures to identify welfare issues as soon as possible. 

Our recommendations cannot replace an HBA, but they can prepare the ethical decision-making process optimally, in addition to a well-designed observation sheet and calculation of animal numbers. 

We hope to trigger a debate on the issues associated with severity assessment and classification of new GA lines, as more research is needed to ensure good welfare for these animals.

## Figures and Tables

**Figure 1 animals-10-00857-f001:**
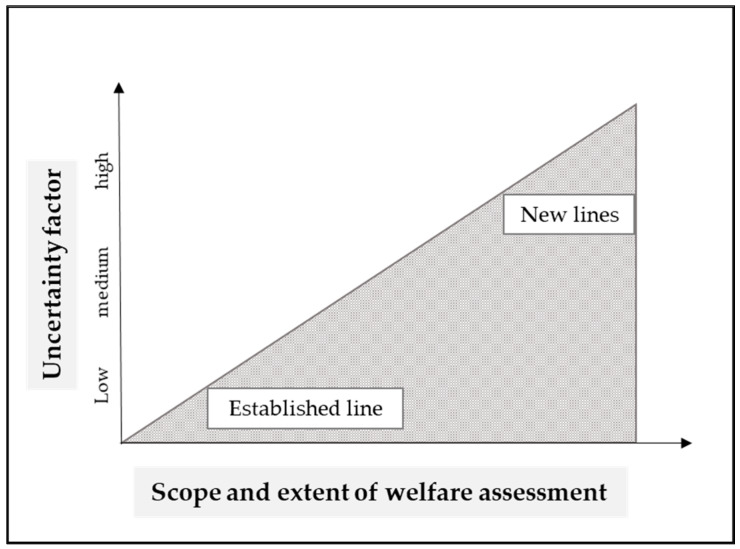
Correlation between uncertainty factor and welfare assessment. The scope and extent of the systematic welfare assessment and measures depend on the uncertainty factor assigned.

**Figure 2 animals-10-00857-f002:**
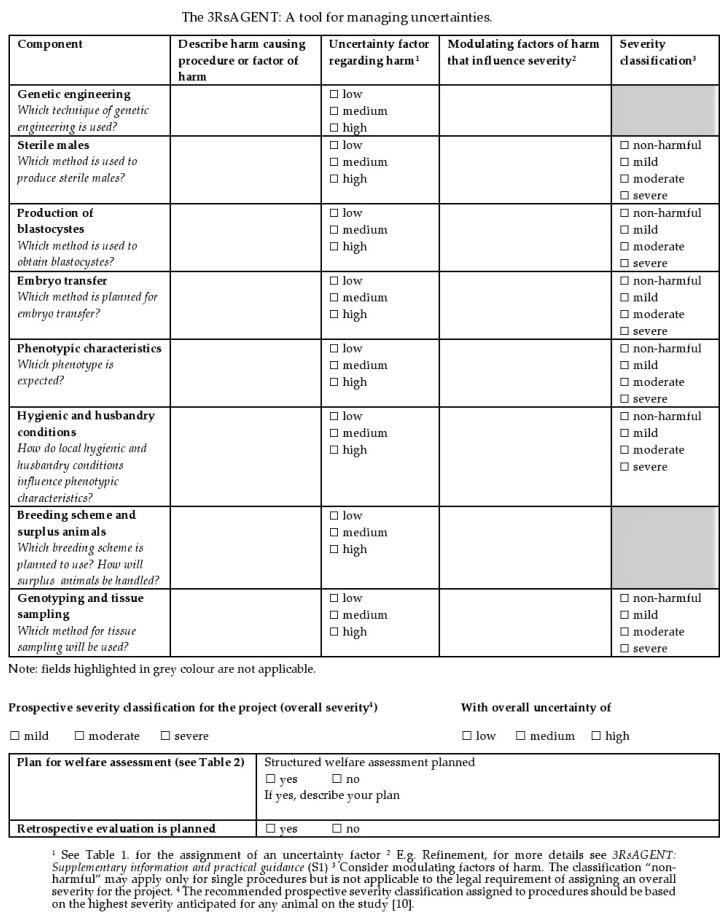
The 3RsAGENT: A tool for managing uncertainties.

**Table 1 animals-10-00857-t001:** Assigning an uncertainty factor.

Uncertainty Factor	Definition	Prospective Severity Classification	Retrospective Assessment
Low	Extensive literature (e.g., scientific articles or databases of genetically altered (GA) animals) or experience with the method or phenotype is available that provides a clear picture of what is expected	Possible	Recommended but not required
Medium	Some literature or experience with the method or phenotype is available that provide indicators for severity assessment, but is still vague	Possible, but vague	Strongly recommended to confirm or revise the prospective severity assessment
High	No literature or experience with the method or phenotype is available	Not possible	Retrospective assessment mandatory

**Table 2 animals-10-00857-t002:** Consequences of uncertainty: How to plan a structured welfare assessment and appropriate monitoring.

Uncertainty Factor	Welfare Assessment Factors (1) Time Points For Assessment	(2) Assessment/Examination Parameters ^1^	(3) Assessors
Low	Animals are assessed at defined time points, from onset of disease until the end of lifespan; frequency according to expected phenotype, e.g., progression of disease.	Adapt the general welfare assessment scheme according to the clinical signs expected.	Scientist and/or animal caretakers; involve veterinarian or Animal Welfare Officer (AWO) if needed.
Medium	Animals are assessed at time points according to expected onset of disease until the end of lifespan; additional time points before and between expected time points should be defined to recognize unexpected phenotypes (all stages of life course should be covered)	Adapt the general welfare assessment scheme according to the clinical signs expected. Include general welfare criteria, e.g., measurement of body weight to recognize unexpected events as soon as possible, and postmortem examination.	Scientist and/or animal caretakers; involve veterinarian or AWO if unexpected phenotypes occur and for the final assessment of the line. Discuss refinement and degree of severity with veterinarian or AWO.
High	All stages of life course should be covered.	Use the general welfare assessment scheme. Include postmortem examination.	Assessment should be carried out by two experienced persons; involve veterinarian or AWO for ongoing monitoring and for the final assessment of the line. Discuss refinement and degree of severity with veterinarian or AWO.

^1^ Assessment should be based on observational parameters and should not involve interventions that may cause additional pain, suffering, or distress. If the characterization of severity of a phenotype requires invasive methods, that should be covered under project authorization.

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
