# Peer review of "Navigating Uncertainties: How to Assess Welfare and Harm in Genetically Altered Animals Responsibly—A Practical Guideline"

_animals, 2020, doi:10.3390/ani10050857_

Round 1

Reviewer 1 Report

The paper "Navigating Uncertainties: How to assess welfare and
harm in GA animals responsibly? A practical guideline " aims to propose the 3RsAGENT (3Rs Assessment of GENetically altered animals - Tool) as a tool to evaluate the chosen methods with respect to defining and best reducing uncertainty or addressing this issue with a caring attitude.
The proposed evaluation tool deserves to be known and applied by the scientific community to improve the evaluation of the welfare of genetically modified animals.
The 3RsAGENT takes into consideration many of the aspects that need to be considered to protect animal welfare.

Based on my professional experience as a veterinarian who is concerned with protecting the welfare of laboratory animals, I would recommend adding a voice regarding the maternal behavior of genetically modified animals. In fact, many strains show not only a reduced fertility, which is mentioned in the paper, but also a poor maternal behavior that often causes the death of most newborns. This fact is certainly the cause of harm that must be taken into account in assessing the seriousness of the procedures for creating a new strain of genetically modified animals.
This is the main suggestion that I feel to give to the authors.
I would also suggest mentioning the following papers in the introduction: David J. Menor-Campos et al. "Attitudes toward Animals of Students at Three European Veterinary Medicine Schools in Italy and Spain". ANTHROZOÖS 2019; 32: 375-385 and Mariti & Gazzano "Refinement of laboratory animal welfare" BIOMEDICAL SCIENCE AND ENGINEERING 2020; 3: 21-22.
Finally I suggest deleting the abbreviation GA from the title and inserting a legend of the abbreviations used before the introduction

Author Response

Comment #1: Based on my professional experience as a veterinarian who is concerned with protecting the welfare of laboratory animals, I would recommend adding a voice regarding the maternal behavior of genetically modified animals. In fact, many strains show not only a reduced fertility, which is mentioned in the paper, but also a poor maternal behavior that often causes the death of most newborns. This fact is certainly the cause of harm that must be taken into account in assessing the seriousness of the procedures for creating a new strain of genetically modified animals.
This is the main suggestion that I feel to give to the authors.

Response #1: Thank you very much for this important note. We agree that a poor maternal behavior presents a serious welfare issue for the offspring and should be considered when evaluating animal welfare concerns of new GA strains. We have integrated an appropriate hint to the Appendix B – 3RsAGENT, Section 5. Phenotype characteristics (Line 612-616).

“It is important to pay good attention on all phenotypic characteristics regardless of whether they affect organic functions or behavioural patterns. There are numerous studies reporting poor maternal behaviour with subsequent negative consequences for the offspring in genetically altered mice [59]. It is obligatory to consider such factors of harm when performing a systematic actual welfare and severity assessment especially in new lines.”

  1. Kuroda, K.O.; Tachikawa, K.; Yoshida, S.; Tsuneoka, Y.; Numan, M. Neuromolecular basis of parental behavior in laboratory mice and rats: with special emphasis on technical issues of using mouse genetics. Prog. Neuropsychopharmacol. Biol. Psychiatry 2011, 35, 1205–1231.

Comment #2: I would also suggest mentioning the following papers in the introduction:

  1. a) David J. Menor-Campos et al. "Attitudes toward Animals of Students at Three European Veterinary Medicine Schools in Italy and Spain". ANTHROZOÖS 2019; 32: 375-385

Response #2a: Thank you for pointing out this reference. Veterinarians have a crucial role in the process of severity assessment and their knowledge and attitude should be seen as driving force to support a thorough severity assessment at research institutes and authorities alike. We agree, that an appropriate training at universities and further training to challenge their attitudes is fundamental for the evaluation process of animal experiments. Nevertheless, our work focuses on the severity assessment process of GA animals. Discussing the different roles of personal involved in this process would go beyond the scope of the paper. Therefore, we would like to exclude the topic from this work and hope for your understanding.

  1. b) Mariti & Gazzano "Refinement of laboratory animal welfare" BIOMEDICAL SCIENCE AND ENGINEERING 2020; 3: 21-22.

Response #2b: Thank you for bringing the reference to our attention. We pointed out in the introduction (line 94 -96) the importance to take all interventions into account and implement stress-reducing measures at an early stage.

“It is therefore important to consider all interventions to ensure animal welfare and to implement refined procedures as early as possible [9].”

  1. Gazzano, A.; Mariti, C. Refinement of laboratory animal welfare. Biomed Sci Eng 2020, doi:10.4081/bse.2019.91.

A more detailed discussion on this topic can be found in sections 2.1 and 2.3.

Comment #3: Finally I suggest deleting the abbreviation GA from the title and inserting a legend of the abbreviations used before the introduction.

Response #3: Thank you for your suggestion. We will waive the abbreviation in the title and revised the title as following: Navigating Uncertainties: How to assess welfare and harm in genetically altered animals responsibly? A practical guideline.

Furthermore, the abbreviations are explained in parentheses when they are used for the first time. We would like to refrain from providing an abbreviation list prior to the introduction as a matter of style of the manuscript and as the format of presentation is in line with the journals submission guidelines. Hence, if the editors would prefer a separate list of abbreviations, we are pleased to provide this list.  

Reviewer 2 Report

  Harm should be singular through out the manuscript.

51 For 2017 should be in 2017

56 harmonized should be coordinate

67 omit to after answer he definition of objective and subjective seems to be reversed here

147 taken into consideration 175 refinement?

201 and 218   born? Should that be living

205 Mendelian capitalize

267-170- These two sentences seem to say the same thing. Perhaps removing “in contrast” will make it clearer

430 until today should be up to the present time

451 do British guidelines apply to the EU anymore?

Cervical dislocation is not humane!

Example 1 hygienic conditions do not influence phenotype

 Example 2. Lactating mice are fed with high-energy nutritional supplement. Why is this considered an endpoint

Table 1 breeding scheme and surplus animals

571 prepubescent FEMALE mice?

600 animals born

649 or the focus

658 is planned to be used

Author Response

Comment #1: Harm should be singular through out the manuscript.

Response #1: Thank you for pointing out the correct wording. We have changed it accordingly in the entire manuscript.

Comment #2: 51 For 2017 should be in 2017

Response #2: Thank you. We implemented the correct grammatical form.

Comment #3: 56 harmonized should be coordinate

Response #3: Thank you for pointing out the clarification. We have changed the word accordingly.

Comment #4: 67 omit to after answer he definition of objective and subjective seems to be reversed here

Response #4: Thank you for this comment. If we understand the comment correctly, you are concerned with the correct sentence structure and the core statement. Therefore, we have revised the sentence to clarify the statement.

“Providing information about the nature and severity of the animals’ harm, pain, suffering, and distress is a legal requirement and serves as an important aspect of the project evaluation. Nevertheless, authorities and applicants still face the challenge to fulfill this legal obligation and to provide sufficient information on projects of GA lines that are likely to develop a harmful phenotype.”

Comment #5: 147 taken into consideration

Response #5: Thank you. The correction has been implemented (now line 155)

Comment #6: 175 refinement?

Response #6: If we understand your comment correctly, your question refers to whether the word “refinement” is used correctly in this context. To clarify the sentence, we have changed the wording to “stress-reducing measures” (now line 183): “Elements to include in an individual severity assessment are the type of technique, species, strain, stage of development, previous experience, frequency, intensity, duration of effect, and effectiveness of stress-reducing measures.”

Comment #7: 201 and 218   born? Should that be living

Response #7: Thank you for pointing this out. We discussed if “born” should be replaced by “living” and decided to generalize the statement and omit “born” or “living”. The reason is that we would also like to include animals that are born dead and include this information in the overall welfare assessment. Therefore, we rephrased the two sentences to “the animals” (now line 210 and 227).

Comment #8: 205 Mendelian capitalize

Response #8: Thank you for indicating the spelling mistake. We have capitalized Mendelian (now line 214).

Comment #9: 267-170- These two sentences seem to say the same thing. Perhaps removing “in contrast” will make it clearer

Response #9: Thank you for pointing out the ambiguous logic of the consecutive sentences. To clarify the statement of this section, we rephrased both sentences (now 276 – 278):

“If the function of the gene is already known or the genetic alteration of a new line suggests a similarity to already existing lines, this information can be used to hypothesize a severity degree. Correspondingly, if knowledge about the expected phenotype is available for established lines, a severity degree, including efficient refinement measures, can be identified prospectively.”

Comment #10: 430 until today should be up to the present time

Response #10: Thank you. The correction has been implemented (now line 439/440).

Comment #11: 451 do British guidelines apply to the EU anymore? (now line 461)

Response #11: Thank you for this question. With this exemplary compilation, we refer to guidelines from different countries regardless if these countries are part of the EU. In our opinion, they provide a valuable source for the scientific community on best practice of information exchange. Up to our knowledge, the guidelines are not legally binding. Therefore, we would like to encourage an international view on exchange of information and also taking guidelines from Switzerland or Greatbritain into account.

Comment #12: Cervical dislocation is not humane!

Response #12: Thank you for this comment. We agree that the killing method of cervical dislocation is under discussion in the scientific community and concerns have been raised due to unsuccessful euthanasia (e.g. Carbone 2011). On the other hand, this killing method is widely used in research institutes and is also allowed according to Annex IV of the Directive 2010/63/EU. A prerequisite for this is, of course, that the staff is well trained in this technique. Nevertheless, we decided to change the method of killing to an overdose of anesthetics according to good veterinary practice. We revised the example as following: 

“Mice are humanely killed by an overdose of anesthetics and blastocystes are harvested.” (see table line 517)

Comment #13: Example 1 hygienic conditions do not influence phenotype

Response #13: Thank you for indicating the grammatical error. We have corrected the sentence accordingly. (See table line 517)

Comment #14:  Example 2. Lactating mice are fed with high-energy nutritional supplement. Why is this considered an endpoint

Response #14: This might be a misunderstanding. Feeding with a high-energy nutritional supplement is meant as supporting lactating mice with additional feed. In order to clarify the statement and distinguish it from the endpoint criteria, we have prefixed "supporting measure”. The table now reads:

“Endpoints: 20% body weight loss (correction with tumor weight) or Body Condition Score 2, size or location of tumors interferes with the ability to move, tumor volume more than 1500 mm3, reduced general health condition.

Supporting measure: Lactating mice are fed with high-energy nutritional supplement.” (See table line 532)

Comment #15: Table 1 breeding scheme and surplus animals

Response #15: I am afraid that we do not unserstand your comment. Please ellaborate your point of critique.

Comment #16: 571 prepubescent FEMALE mice?

Response #16: Thank you for pointing out this specification. We have rephrased “mice” to “female mice” (now line 581).

Comment #17: 600 animals born

Response #17: Thank you. We have changed the order of words to “animals born” (now line 611).

Comment #18: 649 or the focus

Response #18: Thank you. We have corrected the English phrase to “or the focus” (now line 660).

Comment #19: 658 is planned to be used

Response #19: Thank you for pointing out the grammatical error. We have corrected the question accordingly: “Which breeding scheme is planned to be used?” (now table line 673).